# Testosterone-Associated Dietary Pattern Predicts Low Testosterone Levels and Hypogonadism

**DOI:** 10.3390/nu10111786

**Published:** 2018-11-16

**Authors:** Tzu-Yu Hu, Yi Chun Chen, Pei Lin, Chun-Kuang Shih, Chyi-Huey Bai, Kuo-Ching Yuan, Shin-Yng Lee, Jung-Su Chang

**Affiliations:** 1School of Nutrition and Health Sciences, College of Nutrition, Taipei Medical University, Taipei 11031, Taiwan; ma07106003@tmu.edu.tw (T.-Y.H.); yichun@tmu.edu.tw (Y.C.C.); peilin.nhs@gmail.com (P.L.); ckshih@tmu.edu.tw (C.-K.S.); shinyng90@gmail.com (S.-Y.L.); 2Department of Public Health, College of Medicine, Taipei Medical University, Taipei 11031, Taiwan; baich@tmu.edu.tw; 3Department of Public Health, College of Public Health, Taipei Medical University, Taipei 11031, Taiwan; 4Department of Emergency and Critical Care Medicine, Taipei Medical University Hospital, Taipei 11031, Taiwan; traumayuan@gmail.com; 5Graduate Institute of Metabolism and Obesity Sciences, College of Nutrition, Taipei Medical University, Taipei 11031, Taiwan; 6Nutrition Research Center, Taipei Medical University Hospital, Taipei 11031, Taiwan; 7Chinese Taipei Society for the Study of Obesity, CTSSO, Taipei 11031, Taiwan

**Keywords:** testosterone, dietary pattern, iron, red blood cell aggregation, insulin, obesity, hypogonadism, eating out

## Abstract

Obesity and low serum testosterone (T) levels are interrelated and strongly influenced by dietary factors, and their alteration entails a great risk of hypogonadism. Substantial evidence suggests a bidirectional relationship between nutrient metabolism (e.g., glucose, lipids, and iron) and T levels in men; however, T-related dietary patterns remain unclear. This study investigated the dietary patterns associated with serum total T levels and its predictive effect on hypogonadism and the body composition. Anthropometry, blood biochemistry, and food frequency questionnaires were collected for 125 adult men. Dietary patterns were derived using a reduced rank regression from 32 food groups. Overall prevalence rates of central obesity and hypogonadism were 48.0% and 15.7%, respectively. An adjusted linear regression showed that age, insulin, red blood cell (RBC) aggregation, and transferrin saturation independently predicted serum total T levels (all *p* < 0.01). The total T-related dietary pattern (a high consumption of bread and pastries, dairy products, and desserts, eating out, and a low intake of homemade foods, noodles, and dark green vegetables) independently predicted hypogonadism (odds ratio: 5.72; 95% confidence interval: 1.11‒29.51, *p* < 0.05) for those with the highest dietary pattern scores (Q4) compared to those with the lowest (Q1). Scores were also negatively correlated with the skeletal muscle mass (*p* for trend = 0.002) but positively correlated with the total body fat mass (*p* for trend = 0.002), visceral fat mass (*p* for trend = 0.001), and to a lesser extent, subcutaneous fat mass (*p* for trend = 0.035) after adjusting for age. Randomized controlled trials are needed to confirm that improvement in dietary pattern can improve T levels and reduce hypogonadism.

## 1. Introduction

Testosterone (T) is the most important male sex hormone in men and plays key roles in reproductive and sexual functions [1]. T is also involved in erythropoiesis, bone and muscle formation, the body composition, and iron metabolism. In mammals, >95% of T is synthesized by Leydig’s cells of the testes and is regulated through the hypothalamic-pituitary-testicular (HPT) axis [2]. A small fraction of T is derived from the ovaries or the adrenal gland. It was hypothesized that adipose tissue-derived aromatase can convert T to estradiol, which in turn suppresses the release of gonadotropin-releasing hormone (GnRH)-luteinizing hormone (LH) from the hypothalamus [3]. Once it enters the circulation, a large proportion (~66%) of T is tightly bound to sex hormone-binding globulin (SHBG) and approximately 33% of T is bound to albumin with lower binding affinities [4]. Only l‒2% is unbound or free T [4].

Several epidemiological studies demonstrated an inverse relationship between obesity and circulating total T concentrations [5,6,7,8]. Low circulating T levels are also associated with obesity-related cardiometabolic diseases such as metabolic syndrome (MetS) [9], non-alcoholic fatty liver disease (NAFLD) [10], and insulin resistance (IR) [11,12,13]. A meta-analysis that examined 12 studies concluded that low T levels may be associated with increased risks of all-cause and cardiovascular disease-related death, but results also showed considerable between-study heterogeneity [14]. The obesity‒hypogonadism relationship is thought to be bidirectional, and changes in adiposity seem to have greater effects on the HPT axis than hypogonadism has on adiposity or body weight [6,15]. Increased adipokines, such as leptin, and proinflammatory cytokines lead to suppression of the HPT axis [15]. Although a wealth of observational studies showed an association between low T levels and IR [11,12,13], the direction of the causal relationship remains undefined [16]. Visceral fat or adipokines may act as important intermediates in the relationship between IR and hypogonadism. Insulin promotes fat storage and central obesity while sex hormones (e.g., T) stimulate lipolysis [17]. It was shown that moderate obesity decreases total T, largely due to the IR-associated reduction in SHBG [6,18]. T is known to affect the body composition, and T supplementation improves the body composition by increasing the lean body mass and decreasing the fat mass [19,20]. Increased visceral fat mass may promote IR through modulating levels of the insulin sensitizer, adiponectin, and also suppress the HPT axis leading to lower circulating T and SHBG levels [16].

Obesity also affects iron metabolism [21], and iron dysregulation may further contribute to low circulating total T levels [22,23]. Hepcidin is a liver- and adipocyte-secreted hormone and is sensitive to signals of obesity-related inflammation (e.g., interleukin-6 and leptin). Under physiological circumstances, hepcidin controls iron homeostasis through regulating the iron absorption rate and intracellular iron efflux [21]. Obesity-related inflammation triggers hepcidin overproduction, leading to a lower iron absorption rate and impaired tissue iron efflux. Hence, obese men are frequently associated with iron dysregulation (indicated by high hepcidin and tissue iron levels and normal/low levels of serum iron levels or transferrin saturation (TS)) [21]. Increased hepatic iron concentrations are associated with moderate hypogonadotropic hypogonadism [23]. Elevated serum ferritin (SF) predicts low total T levels in Chinese adults [24] and young adolescent Taiwanese males [22].

Accumulating evidence suggests a tight link between the nutritional status and male reproductive function, particularly the effects of calories or macronutrients on male sex hormone total T levels [1]. A 10-day fast in obese men resulted in a significant fall in serum T levels, regardless of whether there was carbohydrate (CHO) supplementation (15 or 45 g/day), and T levels increased to normal during the re-feeding (1500 Kcal/day) period [25]. Malnutrition (e.g., protein restriction or a protein-energy deficiency) was suggested to impair Leydig’s cell function and affect T biosynthesis [26,27]. Human and animal studies also showed that a high fat diet (HFD) is inversely associated with total T levels [28,29,30]; however, conflicting data exist as to the types of fatty acids [28,30,31]. CHO intake may also affect the male sex hormone. A low-CHO diet (<5% of total energy content) decreased plasma total T levels while circulating levels of adrenaline, noradrenaline, and growth hormone increased [32]. An early study by Anderson et al. showed that a high-CHO diet increased circulating total T and SHBG levels, while a high-protein diet had reversed the effect [33]. However, Mikulski et al. showed that both low (35% protein, 64% fat, 1% CHO) and high (4% protein, 1% fat, 95% CHO) CHO meals decreased serum T levels in physically active subjects [34]. Although CHOs seem to be positively correlated with circulating total T and SHBG levels in men [28], increased intake of refined CHO is associated with low serum SHBG levels in both men [16] and women [35]. A recent study showed no significant relationship between dietary intake of CHO and total and free T levels in healthy women [36]. Overall, the effect of CHO intake on androgen may differ between genders.

We hypothesized that obesity might affect not only the male sex hormone, T, but also nutrient metabolism (e.g., iron and glucose), and alterations in nutrient metabolism may contribute to the bidirectional relationship between obesity and the male sex hormone, T. Therefore, the broad aims of this study were to (1) investigate the relationship between serum nutrient biomarkers (e.g., iron and glucose) and total T levels and (2) identify dietary patters associated with circulating total T and its predictive effect on hypogonadism and body composition in 125 Taiwanese male adults.

## 2. Materials and Methods

### 2.1. Participants

This was a cross-sectional study with 125 Taiwanese males aged 20‒64 years. Participants were recruited at the Division of Gastroenterology and Hepatobiliary Diseases, Department of Internal Medicine, Taipei Medical University (TMU) Hospital from 1 June 2015 to 31 September 2015. The sampling method of this study was non-probability volunteer sampling. Participants were excluded from the study if they had any one of the following: (1) a disease history of hepatitis virus B or hepatitis virus A infection; (2) a cholecystectomy or drug-induced hepatitis; (3) excessive alcohol intake, defined by alcohol intake of >30 g/week; (4) chronic disease (e.g., hepatocarcinoma, nephritis, cancer, or autoimmune disease); (5) prostate diseases under therapy; (6) under hormone therapy in the past three months; or (7) having taken iron supplements in the past three months. The protocol was reviewed and approved by the institutional ethical review committee of Taipei Medical University (TMU-JIRB 201502018), and written informed consent was obtained from all participants.

### 2.2. Definitions

Anemia was defined as hemoglobin (Hb) of <13 g/dL, and iron overload was defined as SF of >300 ng/mL [37]. Type 2 diabetes was self-reported or with glycated hemoglobin (HbA1c) of >6.5% [38]. Non-alcoholic fatty liver disease (NALFD) was diagnosed by three experienced physicians using transient elastographic ultrasound scans, on either an Aplio 500 ultrasound machine (Toshiba Medical Systems, Tokyo, Japan) or a Xario XG ultrasound machine (Toshiba Medical Systems). The criteria of MetS were based on the modified National Cholesterol Education Program Adult Treatment Panel III for the Asia Pacific [39], in which participants with at least three of the following were classified as having MetS: (1) a waist circumference (WC) in males of ≥90 cm (also defined as central obesity); (2) systolic blood pressure of ≥130 mmHg or diastolic blood pressure of ≥85 mmHg; (3) fasting plasma glucose of ≥100 mg/dL; (4) high-density lipoprotein cholesterol (HDL-C) of <40 mg/dL; and (5) fasting triglyceride (TG) of ≥150 mg/dL. Hypogonadism was defined according to the U.S. Food and Drug Administration recommendation, which was serum total T of <3 ng/mL [40].

### 2.3. Questionnaires

A health history questionnaire was used to collect information of participants on age, sex, nationality, religion, smoking behavior, alcohol consumption, medical history, recent medication (hormone therapy, iron supplementation), and alcohol abuse. A food frequency questionnaire (FFQ) was used to investigate dietary patterns of participants. This FFQ was modified based on the validated FFQ used for the Nutrition and Health Survey in Taiwan [41]. It contained 66 food items, which were categorized into 32 food groups. Five commonly used cooking methods for protein-rich foods and the frequencies of eating outside and homemade food were also included. The frequency response included eight categories: (1) 0‒1 time/week; (2) 2‒3 times/week; (3) 4‒5 times/week; (4) 6‒7 times/week; (5) 8‒10 times/week; (6) 11‒13 times/week; (7) 14‒16 times/week; and (8) ≥17 times/week.

### 2.4. Anthropometric Measurements

The body weight, height, and the ratio of the waist (WC) and hip circumferences (HC) of each participant were recorded. WC was measured at the midpoint between the lower edge of the rib cage and the top of the iliac crest. The HC was measured at the widest portion of the buttocks. The waist-hip (W/H) ratio was calculated by dividing the WC with the HC. The body mass index (BMI) was calculated as the weight divided by the height squared (kg/m^2^). The body composition was assessed by bioelectrical impedance using an direct segmental multi-frequency bioelectrical impedance analytical meter X-SCAN Plus-II analyzer (X-Plus, Jawon, Korea) [42]. Body compositions were divided by the body weight and expressed as the total body fat mass (BFM) (%), skeletal muscle mass (SMM) (%), visceral fat mass (VFM) (%), and subcutaneous fat mass (SFM) (%).

### 2.5. Laboratory Measurements

Blood (15 mL) was drawn from every participant after 8 h of fasting. All blood samples were taken in the morning (09:00‒12:00) to minimize the circadian effects on the male sex hormone [43]. Whole blood was used to analyze RBC rheology (deformability and aggregation). Both RBC deformability and aggregation were analyzed using the RheoScan-AnD300 microfluidic ektacytometer (MicroStar Instruments, Seoul, Korea). RBC deformability was defined as the shear stress required for the one-half of the maximal elongation (SS^1/2^). RBC aggregation was defined as the critical shear stress (CSS), which was the minimum amount of force required to disperse RBCs [44]. Plasma was used to measure fasting glucose. Serum was used to analyze glucose biomarkers, blood lipid biomarkers, liver injury biomarkers, and iron biomarkers. Serum iron was measured by a colorimetric method (Le-Zen Clinical Laboratory, Taipei, Taiwan). Serum transferrin saturation (%TS) was calculated using the formula: (serum iron ÷ total iron-binding capacity (TIBC)) × 100%. Serum free Hb (Immunology Consultants Laboratory, Portland, OR, USA), serum hepcidin (DRG International, Springfield, NJ, USA), and sCD163 (R&D Systems, Minneapolis, MN, USA) were analyzed using enzyme-linked immunosorbent assays (ELISAs) following the manufacturer’s instructions. Total T and sex hormone-binding globulin (SHBG) were measured by an electrochemiluminescence immunoassay (Cobas E601, Roche Diagnostics, Mannheim, Germany). The detection limit of total T was 0.025 ng/mL, and it was 0.350 nmol/L for SHBG. Bioavailable T (bio-T) and free T (free-T) was calculated using the formula: total T = free-T + albumin-bound-T + SHBG-bound-T [45].

### 2.6. Statistical Analysis

Analyses were conducted using IBM SPSS 21 (IBM, Armonk, NY, USA), SAS vers. 9.4 (SAS Institute, Cary, NC, USA), and GraphPad Prism 5 (GraphPad Software, La Jolla, CA, USA). Categorical data are presented as the number (percentage (%)), and continuous data are presented as the mean ± standard deviation (SD). Total T data were divided into quartiles (Qs) using SPSS with Q1 assigned to the smallest value. A general linear model and Chi-squared test were used to analyze the *p* for trend between variables for continuous data and categorical data, respectively. A reduced rank regression (RRR) was implemented to derive total T-associated dietary patterns with the 32 food groups. The FFQ was used as a predictor, and biomarkers determined from the multivariate linear regression analysis were used as responses [46]. Total T-associated dietary patterns were presented with food groups that had factor loadings of ≥0.20 or ≤−0.20. Dietary pattern scores that were derived from each participant represented the sum of food intake variables weighted by the corresponding factor loading. These scores indicated the conformity of food consumption to the total T-associated dietary pattern. A multivariate linear regression analysis was performed to evaluate relationships between dietary pattern sores and RRR responses and potential variables. The directed acyclic graph below explains the conceptual framework of the RRR (Figure 1). Differences were considered significant at *p* values of ≤0.05.

## 3. Results

### 3.1. Total T Levels Are Positively Associated with TS and SMM and Negatively Associated with Age, Obesity-Related Cardiometabolic Diseases, and RBC Rheology

We stratified individuals according to the total T concentrations into quartiles. Table 1 shows that total T had positive trends with the skeletal muscle mass, %TS, HDL, and sex hormone biomarkers (all *p* for trend <0.01). On the other hand, age (*p* for trend <0.05), central obesity (*p* for trend <0.001), visceral fat mass (*p* for trend <0.001), NAFLD (*p* for trend <0.001), MetS (*p* for trend <0.001), hypogonadism (*p* for trend <0.001), IR (*p* for trend <0.001), TG (*p* for trend <0.01), and RBC rheology (*p* for trend <0.05) had negative trends with total T.

### 3.2. Relationship between Serum Total T and Potential Variables

We next investigated potential confounding variables that were associated with the male sex hormone, total T. A multivariate linear regression analysis was used to explore variables that could independently predict serum total T levels. Table 2 shows that after adjusting for covariates, only log-transformed age (β = −1.159; 95% CI: −1.876‒−0.442, *p* = 0.002), log-transformed insulin (β = −0.713; 95% CI: −1.259‒−0.166, *p* = 0.011), log-transformed RBC aggregation (β = −1.025; 95% CI: −2.043‒−0.008, *p* = 0.048), and %TS (β = 0.675; 95% CI: 0.076‒1.274, *p* = 0.048) remained significantly correlated with serum total T levels (model 3).

### 3.3. T-Associated Dietary Pattern Scores by the Reduced Rank Regression (RRR)

T-associated dietary pattern scores were derived by the RRR for determining the predictability of the dietary pattern towards total T and hypogonadism. Response variables were selected based on strong correlations between total T and the independent variables, which were insulin, %TS, and RBC aggregation (all *p* < 0.05; model 3, Table 2). Table 3 shows that food groups of bread and pastries, dairy products, desserts, and eating out were positively correlated with the first dietary pattern scores (factor loadings of ≥0.20). On the other hand, homemade foods, noodles, and dark green vegetables were negatively correlated with dietary pattern scores (factor loadings of ≤−0.20).

Dietary pattern scores was then stratified into quartile levels for the investigation of the relationships between the dietary pattern scores and potential variables. Table 4 shows that age (*p* < 0.05), hypogonadism (*p* < 0.01), NAFLD (*p* < 0.01), MetS (*p* < 0.05), BMI (*p* = 0.001), total body fat mass (*p* = 0.001), visceral fat mass (*p* < 0.001), glucose biomarkers (all *p* < 0.05), TG (*p* < 0.001), and RBC aggregation (*p* = 0.001) had positive trends with dietary pattern scores. A small positive trend between serum ferritin and dietary pattern scores was also found (*p* = 0.052). However, dietary pattern scores had negative trends with HDL-C (*p* < 0.05), skeletal muscle mass (*p* = 0.001), total T (*p* < 0.001), and free T (*p* < 0.001).

We next performed a multivariate linear regression and logistic regression to investigate the predictive effects of T-associated dietary pattern scores on total T levels, hypogonadism, and body composition. Table 5 shows that when compared to the lowest dietary pattern scores [Q1 (Ref)], individuals with the Q3 (−0.852 (−1.500~−0.205), *p* = 0.011; model 2) and Q4 (−0.872 (−1.722~−0.023), *p* = 0.044; model 2) dietary pattern scores were significantly correlated with lower total T levels after adjusting for age and BMI (*p* for trend <0.01; model 2).

The multivariate logistic regression analysis showed that when compared to the lowest dietary pattern scores (Q1 (Ref)), individuals with the highest dietary pattern scores (Q4) had a 5.72-fold (OR: 5.72; 95% CI: 1.11~29.51, *p* < 0.05) higher risk of developing hypogonadism after adjusting for age and BMI (Figure 2).

An age-adjusted linear regression analysis also showed positive trends of T-associated dietary pattern scores with total body fat mass (*p* for trend = 0.002), visceral fat mass (*p* for trend = 0.001), and subcutaneous fat mass (*p* for trend = 0.035) but a negative trend between scores and skeletal muscle mass (*p* for trend = 0.002) (Table 6, age-adjusted).

## 4. Discussion

Our study found that the male sex hormone, total T, is closely linked to age and nutrition-related biomarkers (e.g., insulin, %TS, and RBC aggregation). Total T-associated dietary patterns, which were characterized by high-frequency consumption of bread and pastries, dairy products, and desserts, eating out, and low intake frequency of homemade foods, noodles, and dark green vegetables, were significantly associated with an unhealthy body composition (indicated as decreased skeletal muscle mass and increased visceral fat mass), low total T levels, and hypogonadism (OR: 5.72; 95% CI: 1.11~29.51, *p* < 0.05).

In the present study, homemade food and eating out contributed 31.2% and 6.2%, respectively, to explaining the variation in the RRR-derived total-T-associated dietary pattern. Foods that are not homemade (consumed when eating out) usually have a heavier taste than homemade food and tend to be Western food. In this study, the three food groups, bread and pastries, dairy products, and desserts, can be grouped as Western foods and were positively associated with eating out. Although noodles are refined CHOs, homemade noodles usually have a lighter taste (containing less fat and salt) than noodles consumed in restaurants. Our study also found that adult men who eat at home and prefer noodles as a staple food are also likely to consume dark green vegetables. Eating more dark green leafy vegetables is known to reduce the risk of type II diabetes. Insulin also contributes up to 28.2% of explaining the variation in the RRR-derived total-T-associated dietary pattern (data not shown), the highest among the various responses. Total T contributes 23.5% to explaining the variation in the RRR-derived total-T-associated dietary pattern (data not shown), the second highest among the responses. It is known that total T and insulin have an inverse relationship as they act like antagonists toward adipocytes [17]. A study on adolescent male rats in 2015 reported that a Western-style diet increased fasting insulin levels and induced IR [47]. Our results from Table 4 show that total-T-associated dietary pattern scores had significant positive trends with the total body fat mass, insulin and HOMA-IR (all *p* < 0.001). Increased adiposity or hyperinsulinemia may suppress total T levels [6,48]. T also plays a critical role in energy metabolism. Varlamov et al. suggested that T exerts positive effects on the skeletal muscle mass and local glucose uptake [49]. Therefore, low T levels may cause obesity due to lower muscle energy expenditure.

We also verified the relationship between dietary pattern scores and the selected responses (total T, insulin, %TS, and RBC aggregation). A multivariate linear regression analysis was used to analyze the responses that were independently related to the RRR-derived dietary pattern. A univariate linear regression suggested that total T (β = −0.204; 95% CI: −0.294~−0.114, *p* < 0.001], log RBC aggregation CSS (β = 0.775; 95% CI: 0.208~1.341, *p* < 0.01), and insulin (β = 0.049; 95% CI: 0.029~0.069, *p* < 0.001) were significantly correlated with the first dietary pattern scores. After adjusting for covariates (age and BMI), only total T (β = −0.130; 95% CI: −0.230~−0.029, *p* < 0.05) and insulin (β = 0.037; 95% CI: 0.012~0.062, *p* < 0.01) remained significantly correlated with T-associated dietary pattern scores (data not shown). This suggests that although insulin, %TS, and RBC aggregation were independent predictors of circulating total T, the contributions of %TS and RBC aggregation to total T-associated dietary pattern were weaker and more likely to be influenced by obesity than insulin and total T.

In the current study, insulin, %TS, and RBC aggregation were identified as independent factors that predicted total T levels, with %TS being positively correlated, and insulin and RBC aggregation being negatively correlated. Accumulating evidence suggests that circulating T levels are inversely associated with IR [11,12,13]; however, the mechanisms underlying this association are less well understood. An early study showed that Leydig’s cells expressed insulin and type I insulin growth factor (IGF) receptors and both insulin and IGF stimulate T production in primary Leydig’s cell culture [50]. However, IR may impair the production of T by Leydig’s cells due to organ resistance to insulin’s actions [51]. In our study, almost half of the middle-age men had central obesity but fewer than 10% had type 2 diabetes (8.7%). It is possible that adiposity-related factors may influence insulin’s effects on Leydig’s cell steroidogenesis. For example, leptin, an adipokine secreted by white adipose tissues, inhibits human chorionic gonadotrophin (hCG)-induced T synthesis in adults testes in vitro [52]. Decreased circulating adiponectin levels in obese men may also lead to HPT axis dysfunction or Leydig’s cell resistance to insulin’s actions [16]. Finally, obesity-related inflammation may directly trigger hypogonadism, but this effect is potentially reversible (e.g., >10% weight loss) [15].

One of the novel findings of this study was the inverse relationship between RBC aggregation and male sex hormone total T levels. Dietary factors are known to affect RBC rheology. Our recent studies found that RBC rheology is strongly influenced by iron and an HFD [53,54]. An animal study found that HFD-fed obese rats had increased RBC aggregation, but iron supplementation reduced HFD-induced RBC aggregation [54]. A human study [53] found that individuals with the highest quartile levels of RBC aggregation had a 2.65-fold (1.054~6.638, *p* < 0.05) highest risk of developing hyperlipidemia than those in the lowest quartile. In addition, RBC aggregation/iron-related dietary patterns independently predicted hyperlipidemia (OR: 2.927; 95% CI: 1.109~7.726, *p* < 0.05) and MetS (OR: 3.904; 95% CI: 1.070~140,248, *p* < 0.05) [53]. Currently, how iron modulates RBC rheology remains unclear. It is likely that iron influences RBC function through modulating membrane fluidity or iron-containing antioxidative enzymes (e.g., superoxide dismutase). In this study, we also observed a positive relationship between RBC aggregation and insulin levels. Early studies found that patients with IR had increased RBC aggregation compared to controls [55], and improved glycemic controls via lifestyle interventions (diet and exercise) decreased RBC aggregation in patients with type 2 diabetes [56].

In spite of a strong correlation between total T and RBC aggregation (*r* = −0.419, *p* < 0.001), the regulatory pathways of RBC aggregation on T levels are still unknown. The literature shows that T supplementation caused erythrocytosis, an increase in the number of RBCs, via increased erythropoietin (EPO), but suppressed hepcidin synthesis, resulting in increased levels of hemoglobin and hematocrit [57]. A retrospective study aiming to determine the rate of secondary polycythemia (hematocrit >50%) in 228 men treated with subcutaneously implanted T pellets found that estimated rates of polycythemia at six, 12, and 24 months were 10.4%, 17.3%, and 30.2%, respectively [58]. On the other hand, T may increases RBC susceptibility to hemolysis possibly through osmotic and oxidative hemolysis [59]. Overall, these data suggest that T and the nutritional status directly modulate RBC function, and whether RBC aggregation directly influences T biosynthesis remains unknown. It is likely that RBC functions (aggregation and deformability) may affect T levels through their ability to change shape and flow in microvessels while transporting oxygen and nutrients to Leydig’s cells of the testes.

This study has several limitations. First, the relatively small sample size (*n* = 125) and cross-sectional design limited the study findings. This study was cross-sectional and only associations can be observed, therefore any inferences about causality cannot be made. Second, a bioelectrical impedance analysis device, and not dual-energy X-ray absorptiometry, was employed to determine the body composition because of budget constraints. Third, this study relied on the use of the FFQ for dietary assessment. The FFQ is a useful tool to assess relationships between diet and disease in general populations. However, the FFQ only represents the frequency of the participants’ food intake in the past three months. Thus, the intake of nutrients cannot be determined. Although potential confounders were considered and adjusted for in our study, there are still some potential factors that may influence the relationship between diet and total T levels, including adipokines, and genetic and socioeconomic factors. Fourth, differences in dietary patterns between age and ethnic groups were observed. In this study, >99% of the study participants were young middle-aged men (with a mean age of 41 ± 11.5 years) of Taiwanese nationality and Han Chinese origin. Hence, the identified dietary patterns may need to be verified in other age (e.g., >65 years) and racial (e.g., Caucasian) groups. In addition, the RRR is a method that relies on prior knowledge of biological relevance to select response variables. Hence, using different responses for the RRR analysis will yield different results. Finally, all blood samples were collected in the morning (09:00~12:00) in order to minimize the diurnal variation of sex hormone levels. The effects of circadian rhythms on sex hormone are well documented, and levels of total, free, and bioavailable T were higher in the morning (08:00) than levels measured in the mid- to late afternoon, which yielded average values 30%~35% higher than at 08:00 [43].

## 5. Conclusions

The relationship between the male sex hormone, T, and obesity is complex, and dietary-related factors may serve as important intermediates. The study results suggest that individuals who prefer Western-style food (bread and pastries, dairy products, and desserts), eat out, and eat fewer homemade foods, noodles, and dark green vegetables are more likely to have an unhealthy body composition (e.g., increased visceral fat and decreased skeletal muscle mass) and low serum total T levels, and are likely to develop hypogonadism. Randomized controlled trials are needed to confirm that an improvement in dietary pattern can improve T levels and reduce hypogonadism.

## Figures and Tables

**Figure 1 nutrients-10-01786-f001:**
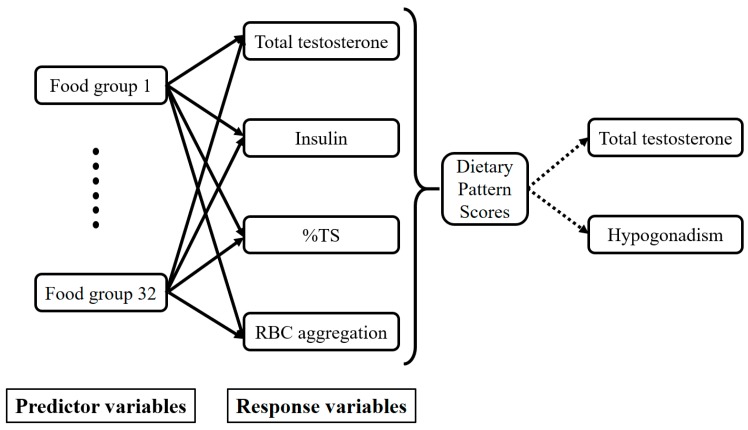
Directed acyclic graph of the reduced rank regression (RRR) conceptual framework. TS: transferrin saturation.

**Figure 2 nutrients-10-01786-f002:**
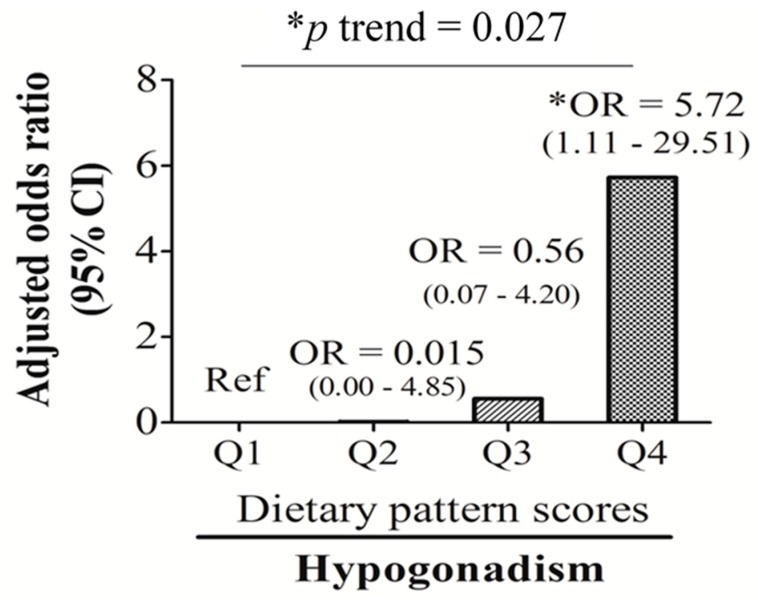
Odds ratios (ORs) and 95% confidence intervals (CIs) of dietary pattern score quartile levels for hypogonadism adjusted by age and log-transformed body-mass index. * *p* ≤ 0.05.

**Table 1 nutrients-10-01786-t001:** Baseline characteristics of the study population according to quartiles of total testosterone levels (*N* = 125).

	Total Testosterone (ng/mL), Quartiles ^$^				*p* for Trend *
	Q1 (*n* = 32)	Q2 (*n* = 30)	Q3 (*n* = 32)	Q4 (*n* = 31)	
Age (years)	44.11 ± 9.89	42.85 ± 10.25	40.37 ± 12.79	37.27 ± 12.55	0.013
Anemia (*n*, %)	1 (3.1)	1 (3.3)	0 (0.0)	1 (3.2)	0.787
TS <20%	8 (25.0)	6 (20.0)	2 (6.3)	1 (3.2)	0.031
Iron overload (*n*, %)	8 (25.0)	12 (40.0)	7 (21.9)	5 (16.1)	0.174
Central obesity (*n*, %)	25 (78.1)	15 (50.0)	13 (40.6)	8 (25.8)	<0.001
Type 2 diabetes (*n*, %)	4 (15.4)	3 (13.6)	2 (7.1)	2 (6.9)	0.653
NAFLD (*n*, %)	31 (96.9)	29 (96.7)	25 (78.1)	16 (51.6)	<0.001
MetS (*n*, %)	12 (46.2)	9 (42.9)	1 (3.6)	4 (13.8)	<0.001
Hypogonadism (*n*, %)	20 (62.5)	0 (0.0)	0 (0.0)	0 (0.0)	<0.001
**Anthropometry**					
W/H ratio	0.93 ± 0.07	0.91 ± 0.04	0.89 ± 0.06	0.87 ± 0.06	<0.001
BMI (kg/m^2^)	28.01 ± 4.10	26.55 ± 4.79	25.36 ± 3.62	23.03 ± 3.20	<0.001
Total body fat mass (%)	27.34 ± 5.04	26.60 ± 4.48	24.94 ± 4.61	22.19 ± 4.82	<0.001
Skeleton muscle mass (%)	66.91 ± 4.90	67.66 ± 4.43	69.32 ± 4.56	72.02 ± 4.78	<0.001
Visceral fat mass (%)	4.40 ± 1.22	4.16 ± 1.23	3.76 ± 1.14	3.12 ± 1.00	<0.001
**Glucose Biomarkers**					
HOMA-IR	3.09 ± 1.85	2.82 ± 1.68	1.60 ± 0.74	1.51 ± 0.62	<0.001
Fasting glucose (mg/dL)	88.78 ± 12.95	96.60 ± 24.17	87.47 ± 8.53	87.52 ± 8.81	0.278
HbA1c (%)	5.89 ± 0.60	5.99 ± 0.97	5.73 ± 0.63	5.61 ± 0.49	0.048
Insulin (µIU/mL)	13.78 ± 6.96	11.58 ± 6.60	7.32 ± 3.13	6.93 ± 2.82	<0.001
**Lipid Biomarkers**					
Total cholesterol (mg/dL)	192.69 ± 40.92	208.13 ± 39.75	201.22 ± 42.58	184.94 ± 26.41	0.322
TG (mg/dL)	173.47 ± 89.90	141.87 ± 56.90	133.22 ± 147.17	96.29 ± 60.22	0.002
HDL-C (mg/dL)	42.90 ± 7.44	48.51 ± 11.32	51.48 ± 13.49	54.93 ± 14.57	<0.001
LDL-C (mg/dL)	119.97 ± 34.55	130.40 ± 36.94	122.03 ± 34.17	106.42 ± 26.84	0.068
**Iron-Related Biomarkers**					
Hb (gm/dL)	15.29 ± 1.76	15.48 ± 2.18	16.51 ± 2.68	15.54 ± 1.09	0.277
Free Hb (µg/mL)	173.59 ± 55.04	144.69 ± 59.00	157.40 ± 45.81	146.03 ± 41.47	0.129
Fe (µg/dL)	99.69 ± 38.27	109.83 ± 40.59	116.31 ± 32.65	115.90 ± 31.98	0.057
Ferritin (ng/mL)	226.75 ± 121.70	260.45 ± 192.24	219.12 ± 141.86	206.27 ± 160.15	0.409
TS (%)	28.76 ± 11.12	32.81 ± 13.21	35.26 ± 12.20	37.06 ± 10.35	0.004
sCD163 (ng/mL)	1050.01 ± 491.15	786.83 ± 338.61	805.03 ± 532.38	791.08 ± 533.29	0.090
Hepcidin (ng/mL)	188.13 ± 102.72	196.78 ± 103.90	198.31 ± 93.59	158.54 ± 84.68	0.286
RBC deformability SS^1/2^ (Pa)	2.28 ± 0.17	2.33 ± 0.19	2.17 ± 0.17	2.20 ± 0.17	0.014
RBC aggregation CSS (mPa)	312.26 ± 64.77	306.70 ± 64.13	260.40 ± 55.98	240.62 ± 44.06	<0.001
**Sex Hormone Biomarkers**					
Total T (ng/mL)	2.57 ± 0.55	3.53 ± 0.19	4.44 ± 0.33	5.80 ± 0.81	<0.001
SHBG (nmol/L)	23.55 ± 12.25	26.87 ± 7.76	34.49 ± 13.21	40.03 ± 14.42	<0.001
Bio-T (ng/mL)	1.52 ± 0.44	1.93 ± 0.31	2.24 ± 0.52	2.80 ± 0.63	<0.001
Free-T (pg/mL)	59.24 ± 16.64	79.10 ± 13.33	88.07 ± 18.31	109.85 ± 23.46	<0.001

* *p* for trend was analyzed by a general linear model for continuous variables and Chi-squared for categorical variables. Continuous data are presented as the mean ± standard deviation, while categorical data are presented as the number (percentage of the same group). ^$^ Total testosterone quartiles: Quartile 1, ≤3.22; Quartile 2, >3.22 and ≤3.83; Quartile 3, >3.83 and ≤4.93; Quartile 4, >4.93 ng/mL. TS, transferrin saturation; NAFLD, non-alcoholic fatty liver disease; MetS, metabolic syndrome; W/H, waist to hip; BMI, body-mass index; HOMA-IR, homeostatic model assessment of insulin resistance; TG, triglyceride, HDL-C, high-density lipoprotein cholesterol; LDL-C, low-density lipoprotein cholesterol; Hb, hemoglobin; Fe, serum iron; sCD163, soluble cluster of differentiation 163; RBC, red blood cell; SS, shear stress; Pa, Pascal; CSS, critical shear stress; mPa, milliPascal; T, testosterone; SHBG, sex hormone-binding globulin; Bio-T bioavailable testosterone.

**Table 2 nutrients-10-01786-t002:** Multivariate linear regression of serum total testosterone levels and selected anthropometric, inflammation, lipid, glucose, and iron-related biomarkers.

Parameters	Univariate		Model 1 *		Model 2 ^#^		Model 3 ^†^	
	β (95% CI)	*p* Value	β (95% CI)	*p* Value	β (95% CI)	*p* Value	β (95% CI)	*p* Value
Log Age (years)	−1.463 (−2.201~−0.724)	<0.001	−1.463 (−2.201~−0.724)	<0.001	−1.098 (−1.903~−0.293)	0.008	−1.159 (−1.876~−0.442)	0.002
W/H ratio	−9.115 (−12.698~−5.531)	<0.001	−7.623 (−11.519~−3.727)	<0.001	−1.610 (−7.150~3.930)	0.566		
Log BMI (kg/m^2^)	−3.789 (−5.057~−2.521)	<0.001	−3.557 (−4.768~−2.346)	<0.001	−3.557 (−4.768~−2.346)	<0.001	−1.426 (−3.104~0.252)	0.095
**Blood Biomarkers**								
HOMA-IR	−0.403 (−0.552~−0.253)	<0.001	−0.366 (−0.511~−0.222)	<0.001	−0.188 (−0.360~−0.016)	0.033		
Fasting glucose (mg/dL)	−0.006 (−0.020~0.007)	0.366						
Log Insulin (µIU/mL)	−1.204 (−1.614~−0.794)	<0.001	−1.156 (−1.543~−0.768)	<0.001	−0.723 (−1.210~−0.235)	0.004	−0.713 (−1.259~−0.166)	0.011
Total cholesterol (mg/dL)	−0.003 (−0.009~0.003)	0.311						
Log TG (mg/dL)	−0.882 (−1.249~−0.516)	<0.001	−0.708 (−1.091~−0.324)	<0.001	−0.345 (−0.732~0.041)	0.080		
Log HDL (mg/dL)	1.908 (0.998~2.817)	<0.001	1.736 (0.865~2.607)	<0.001	0.789 (−0.130~1.707)	0.092		
LDL-C (mg/dL)	−0.006 (−0.013~0.001)	0.059						
RBC (MIL/mm^3^)	0.051 (−0.255~0.357)	0.741						
Log Hb (gm/dL)	1.350 (−0.412~3.111)	0.132						
Free Hb (µg/mL)	−0.002 (−0.007~0.003)	0.350						
Serum Fe (µg/dL)	0.004 (−0.002~0.010)	0.204						
Log Ferritin (ng/mL)	−0.259 (−0.597~0.079)	0.132						
Log TS (%)	0.820 (0.227~1.414)	0.007	0.718 (0.151~1.286)	0.014	0.577 (0.066~1.087)	0.027	0.675 (0.076~1.274)	0.048
sCD163 (ng/mL)	−0.001 (−0.001~−0.000025)	0.062						
Hepcidin (ng/mL)	−0.002 (−0.005~0.001)	0.175						
Log RBC Deformability SS1/2 (Pa)	−3.137 (−6.300~0.025)	0.052						
Log RBC Aggregation CSS (mPa)	−2.447 (−3.483~−1.411)	<0.001	−2.223 (−3.239~−1.207)	<0.001	−1.650 (−2.635~−0.665)	0.001	−1.025 (−2.043~−0.008)	0.048

* Model 1: adjusted for log-age; ^#^ Model 2: adjusted for log-age and log-BMI; ^†^ Model 3: adjusted for log-age, log-BMI, log-insulin, log-TS and log-RBC aggregation. Abbreviations are defined in the footnotes to Table 1.

**Table 3 nutrients-10-01786-t003:** Food groups that were strongly associated with the total testosterone-related dietary pattern scores identified by using a reduced rank regression (RRR).

Food Group	Explained Variation (%)	Factor Loading *
Bread and pastries	13.62	0.35
Dairy products	7.37	0.26
Desserts	6.57	0.24
Eating out	6.30	0.24
Homemade foods	31.17	−0.53
Noodles	10.53	−0.31
Dark green vegetables	7.27	−0.25
Total explained variation (%):	59.53	

* Factor loadings are correlations between food groups and the first dietary pattern scores (correlation coefficient for the RRR-derived pattern ≥|0.20|).

**Table 4 nutrients-10-01786-t004:** Characteristics of the study population according to quartiles of dietary pattern scores.

	Dietary Pattern Scores, Quartile ^$^				*p* for Trend *
	Q1 (*n* = 52)	Q2 (*n* = 53)	Q3 (*n* = 53)	Q4 (*n* = 53)	
Age (years)	37.59 ± 11.81	40.23 ± 11.46	41.78 ± 12.24	45.63 ± 11.24	0.014
Hypogonadism (*n*, %)	3 (11.5)	1 (3.7)	4 (14.8)	10 (38.5)	0.006
Iron overload (*n*, %)	2 (8.0)	7 (25.9)	8 (29.6)	7 (28.0)	0.229
NAFLD (*n*, %)	17 (65.4)	18 (66.7)	26 (96.3)	23 (88.5)	0.008
Type 2 diabetes (*n*, %)	1 (3.8)	1 (3.7)	4 (14.8)	4 (15.4)	0.264
MetS (*n*, %)	2 (8.0)	4 (14.8)	9 (33.3)	11 (44.0)	0.011
**Anthropometry**					
W/H ratio	0.87 ± 0.05	0.90 ± 0.07	0.92 ± 0.05	0.91 ± 0.07	0.007
BMI (kg/m^2^)	23.41 ± 2.22	24.77 ± 4.58	26.65 ± 3.51	26.91 ± 5.22	0.001
Total body fat mass (%)	22.98 ± 4.13	23.72 ± 5.02	26.44 ± 4.28	26.95 ± 6.08	0.001
Skeleton muscle mass (%)	71.48 ± 4.40	70.51 ± 4.99	67.81 ± 4.13	67.32 ± 6.01	0.001
Visceral fat mass (%)	3.25 ± 0.88	3.52 ± 1.18	4.11 ± 0.99	4.35 ± 1.57	<0.001
**Glucose Biomarkers**					
Fasting glucose (mg/dL)	86.64 ± 6.67	87.48 ± 6.71	94.93 ± 26.10	93.92 ± 15.58	0.043
HbA1c (%)	5.56 ± 0.44	5.69 ± 0.60	6.10 ± 1.03	5.98 ± 0.68	0.012
Insulin (µIU/mL)	7.28 ± 3.47	7.01 ± 2.84	11.12 ± 5.32	13.42 ± 7.86	<0.001
HOMA-IR	1.54 ± 0.69	1.52 ± 0.65	2.53 ± 1.14	3.20 ± 2.16	<0.001
**Lipid Biomarkers**					
Total cholesterol (mg/dL)	188.76 ± 26.96	203.96 ± 30.80	200.59 ± 48.19	192.88 ± 49.12	0.802
TG (mg/dL)	85.64 ± 40.81	113.33 ± 54.36	147.44 ± 59.33	160.84 ± 98.34	<0.001
HDL-C (mg/dL)	52.60 ± 11.17	53.51 ± 12.62	47.40 ± 10.79	46.79 ± 15.43	0.038
LDL-C (mg/dL)	113.92 ± 29.42	127.07 ± 29.63	127.89 ± 41.56	115.20 ± 37.50	0.882
**Iron Biomarkers**					
RBC (MIL/mm^3^)	5.32 ± 0.80	5.25 ± 0.71	5.38 ± 0.80	5.23 ± 0.82	0.855
Hb (gm/dL)	15.76 ± 1.97	15.91 ± 2.22	16.04 ± 2.19	15.50 ± 2.33	0.736
Free Hb (µg/mL)	149.03 ± 47.10	147.82 ± 48.55	174.95 ± 53.21	161.42 ± 42.20	0.158
Fe (µg/dL)	108.92 ± 34.95	119.48 ± 29.80	109.00 ± 38.24	111.08 ± 33.85	0.896
Ferritin (ng/mL)	150.09 ± 81.65	248.26 ± 154.48	258.70 ± 191.44	237.62 ± 170.30	0.052
TS (%)	33.14 ± 10.75	36.46 ± 10.48	30.61 ± 12.41	32.64 ± 11.29	0.466
sCD163 (ng/mL)	822.54 ± 523.70	670.10 ± 239.59	940.25 ± 655.14	959.94 ± 437.00	0.144
Hepcidin (ng/mL)	154.22 ± 74.12	175.59 ± 84.87	218.43 ± 112.81	189.68 ± 102.76	0.081
RBC deformability SS^1/2^ (Pa)	2.19 ± 0.18	2.21 ± 0.16	2.25 ± 0.19	2.28 ± 0.19	0.079
RBC aggregation CSS (mPa)	246.59 ± 46.42	260.89 ± 52.52	298.89 ± 68.71	299.50 ± 70.07	0.001
**Sex Hormone Biomarkers**					
Total T (ng/mL)	4.83 ± 1.23	4.73 ± 1.20	3.56 ± 0.83	3.54 ± 1.36	<0.001
Free T (pg/mL)	96.35 ± 21.27	91.31 ± 25.44	79.96 ± 20.97	67.13 ± 23.91	<0.001
SHBG (nmol/L)	32.98 ± 12.64	36.68 ± 15.04	25.03 ± 6.73	35.22 ± 17.21	0.683

* *p* for trend was analyzed by a general linear model for continuous variables and Chi-squared test for categorical variables. Continuous data are presented as the mean ± standard deviation, while categorical data are presented as the number (percentage of the same group). ^$^ Dietary pattern scores quartiles: Quartile 1 ≤−0.45; Quartile 2 >−0.45 to ≤−0.09; Quartile 3 >−0.09 to ≤0.43; Quartile 4 >0.43. NAFLD, non-alcoholic fatty liver disease; MetS, metabolic syndrome; W/H, waist to hip; BMI, body-mass index; HOMA-IR, homeostatic model assessment of insulin resistance; TG, triglyceride, HDL-C, high-density lipoprotein cholesterol; LDL-C, low-density lipoprotein cholesterol; RBC, red blood cell; Hb, hemoglobin; Fe, serum iron; TS, transferrin saturation; sCD163, soluble cluster of differentiation 163; SS, shear stress; Pa, Pascal; CSS, critical shear stress; mPa, milliPascal; T, testosterone; SHBG, sex hormone-binding globulin.

**Table 5 nutrients-10-01786-t005:** Linear regression of the relationship between quartile of dietary pattern score levels and total testosterone.

	Dietary Pattern Scores, Quartiles ^$^							*p* for Trend
	Quartile 1	Quartile 2	*p* Value	Quartile 3	*p* Value	Quartile 4	*p* Value	
Univariate	Ref	−0.105 (−0.784−0.573)	0.756	−1.276 (−1.857–−0.695)	<0.001	−1.297 (−2.035–−0.558)	0.001	<0.001
Model 1 *	Ref	−0.044 (−0.711−0.624)	0.896	−1.198 (−1.778–−0.619)	<0.001	−1.192 (−1.979–−0.406)	0.004	<0.001
Model 2 ^#^	Ref	0.155 (−0.464−0.773)	0.618	−0.852 (−1.500–−0.205)	0.011	−0.872 (−1.722–−0.023)	0.044	0.004

* Model 1: adjusted for age; ^#^ Model 2: adjusted for age and body-mass index. ^$^ Dietary pattern scores quartiles: Quartile 1 ≤−0.45; Quartile 2 >−0.45 to ≤−0.09; Quartile 3 >−0.09 to ≤0.43; Quartile 4 >0.43.

**Table 6 nutrients-10-01786-t006:** Multivariate linear regression of the relationship between quartiles (Qs) of dietary pattern score levels and body composition.

	Dietary Pattern Scores ^$^							*p* for Trend
	Q1	Q2	*p* Value	Q3	*p* Value	Q4	*p* Value	
**Univariate**								
BFM (%)	Ref	0.741 (−1.800~3.283)	0.561	3.456 (1.136~5.776)	0.004	3.969 (1.074~6.864)	0.008	0.001
SMM (%)	Ref	−0.973 (−3.573~1.626)	0.456	−3.673 (−6.027~−1.320)	0.003	−4.162 (−7.095~−1.228)	0.006	0.001
VFM (%)	Ref	0.272 (−0.304~0.849)	0.348	0.862 (0.339~1.384)	0.002	1.098 (0.384~1.812)	0.003	<0.001
SFM (%)	Ref	0.477 (−1.496~2.450)	0.630	2.685 (0.829~4.541)	0.005	8.497 (−2.753~19.747)	0.135	0.030
**Age adjusted**								
BFM (%)	Ref	0.540 (−1.996~3.077)	0.671	3.475 (1.094~5.855)	0.005	3.593 (0.504~6.681)	0.024	0.002
SMM (%)	Ref	−0.763 (−3.356~1.829)	0.557	−3.678 (−6.093~−1.262)	0.004	−3.756 (−6.883~−0.629)	0.020	0.002
VFM (%)	Ref	0.211 (−0.356~0.778)	0.457	0.841 (0.307~1.375)	0.003	0.974 (0.214~1.734)	0.013	0.001
SFM (%)	Ref	0.337 (−1.640~2.314)	0.733	2.708 (0.810~4.607)	0.006	8.637 (−3.453~20.727)	0.157	0.035

^$^ Dietary pattern scores quartiles: Quartile 1 ≤−0.45; Quartile 2 >−0.45 to ≤−0.09; Quartile 3 >−0.09 to ≤0.43; Quartile 4 >0.43. BFM, total body fat mass; SMM, skeletal muscle mass; VFM, visceral fat mass; SFM, subcutaneous fat mass.

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
