# Peer review of "Testosterone-Associated Dietary Pattern Predicts Low Testosterone Levels and Hypogonadism"

_nutrients, 2018, doi:10.3390/nu10111786_

Round 1

Reviewer 1 Report

The topic of this manuscript is of interest, aiming at evaluating the relationship between serum nutrient biomarkers, total testosterone levels and dietary patterns identified by rank regression analysis. However, I have several concerns about this study:

English style and language require extensive English editing. I found several errors, typos and confused paragraphs throughout the text;

Tool for dietary data assessment is not previously validated and - as reported by the Authors - did not collect information on portion sizes. This is the main limitation of this work since reduced rank regression determines linear combinations of predictor variables (e.g., intakes of food groups) that explain as much as possible variation in the response variables (e.g., biomarkers). Using frequency rather than intake of foods is a serious mistake, since foods that are frequently consumed weight more on the dietary patterns than food that are less frequently consumed. Moreover, it is not clear criteria for the classification of FFQ items in food groups. For instance, I did not understand what foods are included in the "eat away from home" group and if these foods are also included in other groups.

Thus, in my opinion, article has serious flaws and research not conducted correctly.

Author Response

Thank you for your commend and we do apologize for the typos and grammatical errors! This manuscript had undergone extensive English editing by a professional English editing service. We totally agree with the major limitation of this study is the lack of portion sizes. This study did not attempt to evaluate the effects of nutrient intake (or portion size) on male sex hormone Testosterone as the role of nutrients (e.g. lipid, carbohydrate, protein) had been intensively evaluated in the early studies. Noethlings U et al also showed that variance in food intake is mainly explained by consumption frequencies rather than portion sizes (J Nutr,2003 ;133(2):510-5; Portion size adds limited information on variance in food intake of participants in the EPIC-Potsdam study.). In addition, there are also publications using frequency rather than the amount of nutrient intakes to assess the dietary pattern by RRR (see table below). The FFQ used in this study was modified based on the validated FFQ used for the Nutrition and Health Survey in Taiwan. We further include the frequencies of eating outside and homemade food in order to understand the life-style factor that may contribute to the male sex hormone. To avoid the confusion, we have changed “eat away from home” to “eat out” throughout the manuscript. Last but not the least, we do appreciate your thoughtful comment on the methodology and we will try to quantitate the food portion size in the future study.

Authors

FFQ as the dietary assessment tools for RRR

quantification of FFQ

Hoffmann et al., 2004 [1]

Self-reported 148 food   item FFQ assessed usual intake during the 12 months before the examination.

P (YES)

Lee et al., 2012 [2]

A FFQ estimated the   average frequency (days/per week) of intake of 21 food groups within the 1   month prior to the interview. Intake frequencies of the 21 food groups were   used as independent variables in RRR analysis for the derivation of dietary   pattern factors.

O (No)

Frank et al., 2015 [3]

A locally specific FFQ   assessed the usual weekly intake frequency of 51 food items in 10 food   categories consumed during the past 12 months with no portion sizes.

O (No)

Biesbroek et al., 2015 [4]

A FFQ containing frequency   and amount of consumption of 178 main food items during the year preceding   enrollment. For each of the 35 food groups, intakes were represented as the   energy percentage they contributed to total energy intake.

P (YES)

Weber et al., 2016 [5]

1.     Habitual food consumption   frequencies were assessed using a qualitative food propensity questionnaire   (FPQ). Transformed standardized consumption frequencies were used as   predictor variables.

2.     Assessment of dietary   intake only covered frequencies and no portion sizes. However, it was previously   shown that variance in food intake is mainly explained by consumption   frequencies rather than portion sizes [6].

O (No)

Lo et al., 2017 [7]

A FFQ on the frequency of   79 food items eaten in the preceding year.

O (No)

Vermeulen et al., 2017 [8]

Four ethnic-specific FFQ   included questions on the portion size and frequency of approximately 200   food items eaten during the past month.

P (YES)

Jacobs et al., 2017 [9]

A validated and calibrated   self-administered quantitative FFQ with >180 food items.

P (YES)

Shin et al., 2018 [10]

A one-day 24-h dietary   recall and a 63-item FFQ designed to assess food and nutrient intakes among Koreans   with 9 intake frequency and 4 serving size questions.

P (YES)

Barroso et al., 2018 [11]

1.     A validated 221 food items   semiquantitative FFQ contained questions on the frequency of consumption over   the previous month and the serving size.

2.     This information was   converted into grams per day consumed, using standardized portion sizes.

P (YES)

Reference

1.           Hoffmann, K.; Schulze, M.B.; Schienkiewitz, A.; Nothlings, U.; Boeing, H. Application of a new statistical method to derive dietary patterns in nutritional epidemiology. American journal of epidemiology 2004, 159, 935-944.

2.           Lee, S.C.; Yang, Y.H.; Chuang, S.Y.; Liu, S.C.; Yang, H.C.; Pan, W.H. Risk of asthma associated with energy-dense but nutrient-poor dietary pattern in taiwanese children. Asia Pac J Clin Nutr 2012, 21, 73-81.

3.           Frank, L.K.; Jannasch, F.; Kroger, J.; Bedu-Addo, G.; Mockenhaupt, F.P.; Schulze, M.B.; Danquah, I. A dietary pattern derived by reduced rank regression is associated with type 2 diabetes in an urban ghanaian population. Nutrients 2015, 7, 5497-5514.

4.           Biesbroek, S.; van der, A.D.; Brosens, M.C.; Beulens, J.W.; Verschuren, W.M.; van der Schouw, Y.T.; Boer, J.M. Identifying cardiovascular risk factor-related dietary patterns with reduced rank regression and random forest in the epic-nl cohort. Am J Clin Nutr 2015, 102, 146-154.

5.           Weber, K.S.; Knebel, B.; Strassburger, K.; Kotzka, J.; Stehle, P.; Szendroedi, J.; Mussig, K.; Buyken, A.E.; Roden, M.; Group, G.D.S. Associations between explorative dietary patterns and serum lipid levels and their interactions with apoa5 and apoe haplotype in patients with recently diagnosed type 2 diabetes. Cardiovascular diabetology 2016, 15, 138.

6.           Noethlings, U.; Hoffmann, K.; Bergmann, M.M.; Boeing, H.; European Investigation into, C.; Nutrition. Portion size adds limited information on variance in food intake of participants in the epic-potsdam study. J Nutr 2003, 133, 510-515.

7.           Lo, Y.L.; Hsieh, Y.T.; Hsu, L.L.; Chuang, S.Y.; Chang, H.Y.; Hsu, C.C.; Chen, C.Y.; Pan, W.H. Dietary pattern associated with frailty: Results from nutrition and health survey in taiwan. Journal of the American Geriatrics Society 2017, 65, 2009-2015.

8.          Vermeulen, E.; Stronks, K.; Snijder, M.B.; Schene, A.H.; Lok, A.; de Vries, J.H.; Visser, M.; Brouwer, I.A.; Nicolaou, M. A combined high-sugar and high-saturated-fat dietary pattern is associated with more depressive symptoms in a multi-ethnic population: The helius (healthy life in an urban setting) study. Public health nutrition 2017, 20, 2374-2382.

9.           Jacobs, S.; Kroeger, J.; Schulze, M.B.; Frank, L.K.; Franke, A.A.; Cheng, I.; Monroe, K.R.; Haiman, C.A.; Kolonel, L.N.; Wilkens, L.R., et al. Dietary patterns derived by reduced rank regression are inversely associated with type 2 diabetes risk across 5 ethnic groups in the multiethnic cohort. Curr Dev Nutr 2017, 1, e000620.

10.         Shin, D.; Lee, K.W.; Kim, M.H.; Kim, H.J.; An, Y.S.; Chung, H.K. Identifying dietary patterns associated with mild cognitive impairment in older korean adults using reduced rank regression. Int J Environ Res Public Health 2018, 15.

11.         Barroso, M.; Beth, S.A.; Voortman, T.; Jaddoe, V.W.V.; van Zelm, M.C.; Moll, H.A.; Kiefte-de Jong, J.C. Dietary patterns after the weaning and lactation period are associated with celiac disease autoimmunity in children. Gastroenterology 2018, 154, 2087-2096 e2087.

Reviewer 2 Report

Title:Testosterone-Associated dietary pattern predicts low testosterone levels and hypogonadism

Authors: Tzu-Yu Hu, Yi-Chun Chen, Pei Lin, Chun-Kuang Shih, Chyi-Huey Bai, Kuo-Ching Yuan, Shin-Ying Lee, Jung-Su Chang

Manuscript#:nutrients-386370

Summary:This manuscript focuses on investigating the dietary pattern that is associated with serum total testosterone levels and its predictive effects on hypogonadism and body composition. Data generated from 125 adult men shows that age, insulin, red blood cells aggregation and transferrin saturation predicted serum total testosterone levels. The higher quartile was positively corelated with total body mass. This study could be important to the dieticians to develop a life-style and food regimens that can be used to prevent obesity-related male sex hormone dysfunction.

Critique: This study is important in understanding correlation between dietary intake on testosterone levels. They have very well addressed the limitations of the current study and what can be done in future to address that in the discussion section.

Author Response

Thank you so much for your kind commend.

Reviewer 3 Report

The study is interesting, however, the most important drawback of the study concer the lacking of results of the food frequency questionnaire used to investigate dietary patterns of participants. It is necessary and useful to the reader to know the quantitative and qualitative food intake. As cited in the title, the effect of dietary pattern on testosterone secretion is of great importance. Table 3 summarizes food groups with the explained variation, but it is not useful to undestand any correlation between calorie, protein, carbohydrates and lipis ingestion with testosterone level and other variables such as free T (or Bio T), insulin resistance, HOMA, iron level,etc. (see also Bianchi VE et al. Obes Rev. 2018 ).

Line 70. T is involved in many other function, please complete with references. 

Line 83-90. This paragraph should be rewritten to better explain the effect of  catbohydrates, fats and protein ingestion on testosterone secretion. (Mikulski T, Ziemba A, Nazar K.J Physiol Pharmacol. 2010) (Kyung NH et al. J Clin Endocrinol Metab.1985) and the effect of carbohudrates intake has a different effect in men and in women (Sjaarda LA,et al. J Clin Endocrinol Metab. 2015)

Author Response

Thank you for your commend. With regards to the specific comments: (1) we have added this reference (Bianchi VE et al. Obes Rev. 2018) to the sentence in the line 70 (page 2, lines 73). (2) We have revised introduction in Line 83-90 (4th paragraph) by incorporating references of Mikulski T et al, Kyung NH et al. and Sjaarda LA et al. (page 2, lines 87-104).

We totally agree with the major limitation of this study is the lack of portion sizes. However, this study did not attempt to evaluate the effects of nutrient intake (or portion size) on male sex hormone Testosterone as the role of nutrients (e.g. lipid, carbohydrate, protein) had been intensively evaluated in the early studies. In addition, Noethlings U et al also showed that variance in food intake is mainly explained by consumption frequencies rather than portion sizes (J Nutr,2003 ;133(2):510-5; Portion size adds limited information on variance in food intake of participants in the EPIC-Potsdam study.). There are also publications using frequency rather than the amount of nutrient intakes to assess the dietary pattern by RRR (see table below). Last but not the least, we do appreciate your thoughtful comment on the methodology and we will try to quantitate the food portion size in the future study.

Authors

FFQ as the dietary assessment tools for RRR

quantification of FFQ

Hoffmann et al., 2004 [1]

Self-reported 148 food   item FFQ assessed usual intake during the 12 months before the examination.

P (YES)

Lee et al., 2012 [2]

A FFQ estimated the   average frequency (days/per week) of intake of 21 food groups within the 1   month prior to the interview. Intake frequencies of the 21 food groups were   used as independent variables in RRR analysis for the derivation of dietary   pattern factors.

O (No)

Frank et al., 2015 [3]

A locally specific FFQ   assessed the usual weekly intake frequency of 51 food items in 10 food   categories consumed during the past 12 months with no portion sizes.

O (No)

Biesbroek et al., 2015 [4]

A FFQ containing frequency   and amount of consumption of 178 main food items during the year preceding   enrollment. For each of the 35 food groups, intakes were represented as the   energy percentage they contributed to total energy intake.

P (YES)

Weber et al., 2016 [5]

1.     Habitual food consumption   frequencies were assessed using a qualitative food propensity questionnaire   (FPQ). Transformed standardized consumption frequencies were used as   predictor variables.

2.     Assessment of dietary   intake only covered frequencies and no portion sizes. However, it was   previously shown that variance in food intake is mainly explained by   consumption frequencies rather than portion sizes [6].

O (No)

Lo et al., 2017 [7]

A FFQ on the frequency of   79 food items eaten in the preceding year.

O (No)

Vermeulen et al., 2017 [8]

Four ethnic-specific FFQ   included questions on the portion size and frequency of approximately 200   food items eaten during the past month.

P (YES)

Jacobs et al., 2017 [9]

A validated and calibrated   self-administered quantitative FFQ with >180 food items.

P (YES)

Shin et al., 2018 [10]

A one-day 24-h dietary   recall and a 63-item FFQ designed to assess food and nutrient intakes among Koreans   with 9 intake frequency and 4 serving size questions.

P (YES)

Barroso et al., 2018 [11]

1.     A validated 221 food items   semiquantitative FFQ contained questions on the frequency of consumption over   the previous month and the serving size.

2.     This information was   converted into grams per day consumed, using standardized portion sizes.

P (YES)

Reference

1.           Hoffmann, K.; Schulze, M.B.; Schienkiewitz, A.; Nothlings, U.; Boeing, H. Application of a new statistical method to derive dietary patterns in nutritional epidemiology. American journal of epidemiology 2004, 159, 935-944.

2.           Lee, S.C.; Yang, Y.H.; Chuang, S.Y.; Liu, S.C.; Yang, H.C.; Pan, W.H. Risk of asthma associated with energy-dense but nutrient-poor dietary pattern in taiwanese children. Asia Pac J Clin Nutr 2012, 21, 73-81.

3.           Frank, L.K.; Jannasch, F.; Kroger, J.; Bedu-Addo, G.; Mockenhaupt, F.P.; Schulze, M.B.; Danquah, I. A dietary pattern derived by reduced rank regression is associated with type 2 diabetes in an urban ghanaian population. Nutrients 2015, 7, 5497-5514.

4.           Biesbroek, S.; van der, A.D.; Brosens, M.C.; Beulens, J.W.; Verschuren, W.M.; van der Schouw, Y.T.; Boer, J.M. Identifying cardiovascular risk factor-related dietary patterns with reduced rank regression and random forest in the epic-nl cohort. Am J Clin Nutr 2015, 102, 146-154.

5.           Weber, K.S.; Knebel, B.; Strassburger, K.; Kotzka, J.; Stehle, P.; Szendroedi, J.; Mussig, K.; Buyken, A.E.; Roden, M.; Group, G.D.S. Associations between explorative dietary patterns and serum lipid levels and their interactions with apoa5 and apoe haplotype in patients with recently diagnosed type 2 diabetes. Cardiovascular diabetology 2016, 15, 138.

6.           Noethlings, U.; Hoffmann, K.; Bergmann, M.M.; Boeing, H.; European Investigation into, C.; Nutrition. Portion size adds limited information on variance in food intake of participants in the epic-potsdam study. J Nutr 2003, 133, 510-515.

7.           Lo, Y.L.; Hsieh, Y.T.; Hsu, L.L.; Chuang, S.Y.; Chang, H.Y.; Hsu, C.C.; Chen, C.Y.; Pan, W.H. Dietary pattern associated with frailty: Results from nutrition and health survey in taiwan. Journal of the American Geriatrics Society 2017, 65, 2009-2015.

8.          Vermeulen, E.; Stronks, K.; Snijder, M.B.; Schene, A.H.; Lok, A.; de Vries, J.H.; Visser, M.; Brouwer, I.A.; Nicolaou, M. A combined high-sugar and high-saturated-fat dietary pattern is associated with more depressive symptoms in a multi-ethnic population: The helius (healthy life in an urban setting) study. Public health nutrition 2017, 20, 2374-2382.

9.           Jacobs, S.; Kroeger, J.; Schulze, M.B.; Frank, L.K.; Franke, A.A.; Cheng, I.; Monroe, K.R.; Haiman, C.A.; Kolonel, L.N.; Wilkens, L.R., et al. Dietary patterns derived by reduced rank regression are inversely associated with type 2 diabetes risk across 5 ethnic groups in the multiethnic cohort. Curr Dev Nutr 2017, 1, e000620.

10.         Shin, D.; Lee, K.W.; Kim, M.H.; Kim, H.J.; An, Y.S.; Chung, H.K. Identifying dietary patterns associated with mild cognitive impairment in older korean adults using reduced rank regression. Int J Environ Res Public Health 2018, 15.

11.         Barroso, M.; Beth, S.A.; Voortman, T.; Jaddoe, V.W.V.; van Zelm, M.C.; Moll, H.A.; Kiefte-de Jong, J.C. Dietary patterns after the weaning and lactation period are associated with celiac disease autoimmunity in children. Gastroenterology 2018, 154, 2087-2096 e2087.

Round 2

Reviewer 1 Report

Not counting my comments on the dietary assessment method, the authors improved their manuscript

Author Response

Thank you for your commend and we do appreciate your commend on the dietary assessment.